# Targeting Phospholipase D Pharmacologically Prevents Phagocytic Function Loss of Retinal Pigment Epithelium Cells Exposed to High Glucose Levels

**DOI:** 10.3390/ijms231911823

**Published:** 2022-10-05

**Authors:** Vicente Bermúdez, Paula Estefanía Tenconi, María Sol Echevarría, Aram Asatrian, Jorgelina Muriel Calandria, Norma María Giusto, Nicolas Guillermo Bazan, Melina Valeria Mateos

**Affiliations:** 1Instituto de Investigaciones Bioquímicas de Bahía Blanca (INIBIBB), Consejo Nacional de Investigaciones Científicas y Técnicas (CONICET), Bahía Blanca 8000, Argentina; 2Departamento de Biología, Bioquímica y Farmacia (DBByF), Universidad Nacional del Sur (UNS), Bahía Blanca 8000, Argentina; 3Neuroscience Center of Excellence, School of Medicine, Louisiana State University Health New Orleans, New Orleans, LA 70112, USA

**Keywords:** phospholipase D (PLD), retinal pigment epithelium (RPE), phagocytosis, inflammation, oxidative stress

## Abstract

We previously described the participation of canonical phospholipase D isoforms (PLD1 and PLD2) in the inflammatory response of retinal pigment epithelium (RPE) cells exposed to high glucose concentrations (HG). Here, we studied the role of the PLD pathway in RPE phagocytic function. For this purpose, ARPE-19 cells were exposed to HG (33 mM) or to normal glucose concentration (NG, 5.5 mM) and phagocytosis was measured using pHrodo™ green bioparticles^®^ or photoreceptor outer segments (POS). HG exposure for 48 and 72 h reduced phagocytic function of ARPE-19 cells, and this loss of function was prevented when cells were treated with 5 μM of PLD1 (VU0359595 or PLD1i) or PLD2 (VU0285655-1 or PLD2i) selective inhibitors. Furthermore, PLD1i and PLD2i did not affect RPE phagocytosis under physiological conditions and prevented oxidative stress induced by HG. In addition, we demonstrated PLD1 and PLD2 expression in ABC cells, a novel human RPE cell line. Under physiological conditions, PLD1i and PLD2i did not affect ABC cell viability, and partial silencing of both PLDs did not affect ABC cell POS phagocytosis. In conclusion, PLD1i and PLD2i prevent the loss of phagocytic function of RPE cells exposed to HG without affecting RPE function or viability under non-inflammatory conditions.

## 1. Introduction

The retina is one of the tissues with the highest metabolic demand in the body, and is exposed to many stressors, such as inflammation and oxidative stress, which are involved in the pathogenesis of several ocular diseases that eventually end in vision loss. Inflammation is a key factor in the pathogenesis of many retinal and ocular diseases, such as age-related macular degeneration (AMD), diabetic retinopathy (DR), retinitis pigmentosa, and uveitis [1,2,3,4]. DR is the main worldwide cause of visual impairment and blindness in working-age adults [5]. In addition, AMD is the leading cause of blindness in elderly people [2,6]. In both conditions, inflammation is a key factor in their pathogenesis, and the development of novel pharmacological therapies can be expected to have a significant impact on public health. For these retinal conditions, pharmacological treatment options are limited to anti-vascular endothelial growth factor (VEGF) drugs to prevent neovascularization and glucocorticoids to reduce the inflammatory response. Therefore, in order to provide new insights for the treatment of ocular inflammatory diseases it is important to elucidate the molecular mechanisms involved in these disorders.

The retinal pigment epithelium (RPE) is a monolayer of pigmented cells placed between the choroid and the photoreceptor (PR) layer. RPE and PR cells are closely interacting partners, with RPE being crucial for the maintenance of visual function [7,8]. Among the functions of RPE are: forming part of the blood–retina barrier; absorption of light energy and protection against photo-oxidation; transport of water, nutrients, and metabolic end products; all-*trans*-retinal to all-*cis*-retinal reisomerization to maintain the visual cycle; secretion of growth factors, immunosuppressive factors, and cytokines; and phagocytosis of PR outer segments (POS) [7,8]. In fact, RPE cells are the most active phagocytic cells in the human body, and defects in the phagocytic process leads to impaired retinal function [7,9,10].

Classical (or canonical) phospholipase D isoforms (PLD1 and PLD2) hydrolyze phosphatidylcholine (PC), generating phosphatidic acid (PA), which can then be dephosphorylated by lipid phosphate phosphatases (LPPs) to yield diacylglycerol (DAG) [11,12,13,14]. Through the generation of the above-mentioned bioactive lipids, PLDs regulate the activity of PA- and DAG-responding proteins such as mTOR (mammalian target of rapamycin) and classical and novel protein kinases C (PKCs), among many others [15,16,17,18]. Moreover, PA and DAG have small head-groups that induce the formation of negative curvature at membranes, and can recruit many proteins involved in cytoskeletal reorganization, with PLD activity thus being crucial for exocytosis, endocytosis, and membrane vesicle trafficking in general [17,19]. Therefore, PLD activity can modulate a wide range of important cellular events. 

Previous findings from our laboratory described for the first time that PLD1 and PLD2 mediate the inflammatory response of human RPE cells. We demonstrated that lipopolysaccharide (LPS) and high glucose levels (HG) can induce the inflammatory response of RPE cells through a mechanism that includes oxidative stress and concatenated activation of PLD1 and 2, extracellular regulated kinase 1/2 (ERK1/2), and nuclear factor kappa B (NFκB), followed by induction of pro-inflammatory interleukins (IL-6 and IL-8) and cyclooxygenase-2 (COX-2) [20,21]. 

Our previous findings postulated for the first time that selective pharmacological PLD1 and PLD2 inhibitors (PLD1i and PLD2i) could be promising therapeutic tools to treat inflammatory events in the retina [20,21,22]. Using specific PLD1i and PLD2i, we were able to prevent the inflammatory response of RPE cells, caspase-3 activation, and the reduction in cell viability induced by HG and LPS [20,21,23]. In addition, we demonstrated that PLD1 and PLD2 play crucial roles in the phagocytic process of macrophages [24,25,26,27,28], although the role of this lipid signaling pathway in RPE phagocytosis has not yet been studied. Therefore, in the present work we aim to study the role of the PLD pathway in the phagocytic function of RPE.

## 2. Results

### 2.1. Inflammatory Inductors Reduce Phagocytic Function of ARPE-19 Cells

Cellular regulation of RPE phagocytosis can be studied in cell cultures by challenging RPE cells (primary culture as well as ARPE-19 cells) with isolated POS [9,29,30,31]. In addition, it has been reported that RPE cells can mediate non-specific phagocytosis by engulfing latex beads, for example [9,29,32]. To study the effect of HG on ARPE-19 cells phagocytic function, they were seeded in 96-well plates and exposed to HG (33 mM) for 24, 48, or 72 h. Then, non-specific phagocytosis was measured using pHrodo™ green E. coli BioParticles^®^, as described in the Materials and Methods section. Because pHrodo™ fluorogenic dye dramatically increases its fluorescence as the pH of its surroundings becomes more acidic, it allows measurement of phagocytic activity based on acidification of the particles as they are ingested. Figure 1A shows that phagocytosis of bioparticles was reduced by 50% and 64% in ARPE-19 cells exposed to HG for 48 h and 72 h, respectively (Figure 1A). Bioparticles phagocytosis was reduced by 25% in ARPE-19 cells exposed to LPS (25 μg/mL) for 48 h (Appendix A).

### 2.2. PLD1 and PLD2 Inhibition Prevents Phagocytic Function Loss of ARPE-19 Cells Exposed to High Glucose Levels, but Does Not Affect Phagocytic Function under Basal Conditions

In order to be able to use pharmacological PLD inhibitors as therapeutic tools to treat or prevent inflammatory events in the retina, it should be demonstrated that these inhibitors do not affect the viability and functionality of retinal cells per se. Previous results from our laboratory have shown that PLD1 and PLD2 activation is an early event in HG-exposed ARPE-19 cells, being observed after 4 h of HG exposure and preceding the expression of proinflammatory mediators (observed after 24 h of HG exposure) and cell viability loss (observed after 72 h of HG exposure) [21]. To study the effect of PLD1i and PLD2i on RPE phagocytic function, ARPE-19 cells were seeded in 96-well plates and pre-incubated with PLD1i or PLD2i (0.5 or 5 μM) for 30 min before HG exposure. PLD inhibitors were present (co-incubated) during HG exposure, and non-specific phagocytosis was measured as previously described. As shown in Figure 1A, RPE phagocytic function was reduced by 50% after 48 h of exposure to HG (33 mM). Figure 1B shows that in ARPE-19 cells exposed to HG and pre- and co-incubated with PLD1i or PLD2i (0.5 and 5 μM), no differences were observed in bioparticle phagocytosis when compared to the control condition. Furthermore, 48 h incubation with 5 μM of PLD1i or PLD2i did not significantly affect ARPE-19 cells bioparticles phagocytosis under non-inflammatory conditions (Figure 1C). To study the effect of HG on PLD1 and PLD2 expression, qPCR assays were performed in ARPE-19 cells exposed to HG for 24 h, an incubation time in which the role of classical PLDs in pro-inflammatory mediators’ expression has been demonstrated but for which cell viability and functionality have not yet been affected [21]. After 24 h HG treatment, a significantly higher mRNA accumulation of PLD1 (34%) and PLD2 (40%) was observed with respect to ARPE-19 cells maintained in NG (Figure 1D).

In addition, we measured POS phagocytosis by ARPE-19 cells exposed to HG for 48 h and incubated or not with PLD inhibitors. For this purpose, cells were exposed to the experimental conditions, then incubated with bovine POS for 16 h. Although it was previously shown that internalized POS are detected in ARPE-19 cells after a 6 h incubation time, POS phagocytosis was shown to be maximal after a 16 h incubation [29]. Then, internalized POS were identified by immunocytochemistry followed by fluorescence and confocal microscopy or by western blot (WB) using an anti-rhodopsin antibody. Because RPE cells do not express rhodopsin, the detection of this protein in these cells is a reliable indicator of POS phagocytosis. Under control conditions, 84% of cells presented phagocyted POS; this percentage was reduced to 57% in ARPE-19 cells exposed to HG for 48 h (Figure 2A). However, in cells exposed to HG and pre- and co-incubated with PLD1i (5 μM) or PLD2i (5 μM), no differences were observed with respect to the control (Figure 2A). Confocal images with orthographic projections showed POS intracellular localization (Figure 2A). When POS phagocytosis was measured by WB, rhodopsin/GAPDH content was reduced by 39% in cells exposed to HG for 48 h with respect to NG control conditions (Figure 2B). However, no differences were observed in rhodopsin/GAPDH content between cells exposed to HG and treated with 5 μM PLD1i or PLD2i and the control condition (Figure 2B). These results demonstrate that PLD1 and PLD2 inhibition prevents phagocytic function loss of ARPE-19 cells exposed to HG.

Our previous results demonstrated increased ROS levels in RPE cells exposed to LPS for 24 h or to HG for 72 h [21,23]. Figure 2C shows that ROS levels were increased after 24 h exposure to HG (by 150%) and that PLD1i and PLD2i (at the lowest concentration used, 0.5 μM) were able to prevent oxidative stress induced by HG exposure in RPE cells.

### 2.3. PLD1 and PLD2 Are Expressed in ABC Cells, a Novel Human RPE Cell Line

Previous findings from our lab reported for the first time the expression of PLD1 and PLD2 in RPE cell lines (ARPE-19 and D407) and their activation in response to HG and LPS [20,21,23]. In the present work, we demonstrate PLD1 and PLD2 expression in a novel human RPE cell line, termed ABC cells. ABC cells, having spontaneously arisen from a primary cell culture in Dr. N. Bazan’s laboratory, present typical characteristics of native RPE cells. They form microvilli, tight junctions, and honeycomb packing. In addition, ABC cells express the right markers compared to ARPE-19 cells and phagocyte POS [29].

Figure 3A, B shows that both canonical PLD isoforms are expressed in ABC cells. We were able to reduce PLD1 expression by 55% by transfecting ABC cells with PLD1 siRNA and by 40% when a combination of PLD1 and PLD2 siRNAs was used (Figure 3A). However, PLD2 expression was reduced by 21% when ABC cells were transfected with the combination of PLD1 and PLD2 siRNAs (Figure 3B). In all the cases, a high transfection efficiency was accomplished (Appendix A). Although we were not able to achieve full silencing of PLDs, the partial reduction in PLD1 and PLD2 expression induced by the combination of both siRNAs was able to reduce by 25% the phosphorylation (activation) of the mTOR downstream effector p70 S6K (Figure 3D), suggesting reduced PA-mediated mTOR activation. RNAseq analysis was performed as previously described [29] in ABC, ARPE-19, and hRPE49 cells. Figure 3C shows that the three RPE cell lines expressed both classical PLDs and that PLD1 and PLD2 gene expression was lower in ARPE-19 and hRPE49 cells with respect to ABC cells (Figure 3C).

Regarding RPE phagocytic function, POS phagocytosis in ABC cells was shown to be maximal after 16 h incubation [29]. Furthermore, ABC cells showed a four-fold increase in opsin internalization compared to ARPE-19 cells, confirming their superior phagocytic properties [29]. PLD1 and PLD2 partial silencing did not affect POS phagocytosis by ABC cells with respect to the control condition (Figure 3E). This result suggests that PLD activity is not crucial to the physiological phagocytic function of RPE under non-inflammatory basal conditions, which is in alignment with the results obtained in ARPE-19 cells treated with PLD1i or PLD2i (Figure 1C).

### 2.4. PLD Pharmacological Inhibitors Do Not Affect ABC Cell Viability under Basal Conditions

Previous results from our lab have shown that treatment with 0.5 μM PLD1i or PLD2i for 48 h does not affect the viability of RPE cells [20]. Furthermore, 48 h incubation with a higher concentration (5 μM) of PLD2i did not affect RPE cells viability either, while the inhibition of PLD1 for 48 h with 5 μM PLD1i reduced cell viability by only 10% [23]. In the present study, we determined that treatment of ABC cells with 5 μM of PLD1i, PLD2i, or their combination did not change their viability, as shown by flow cytometry (Figure 4). These results agree with our previous findings in ARPE-19 and D407 cells.

## 3. Discussion

In the present work, we studied the role of the PLD pathway in RPE phagocytic function. The process of disk shedding and phagocytosis must be tightly coordinated between RPE and PR to assure a proper length of POS [7]. POS of cones and rods undergo constant renewal to maintain excitability of PR, and defects in the phagocytic process lead to impaired retinal function [7,9,10]. To our knowledge, the role of PLD1 and PLD2 in POS phagocytosis has not been previously investigated. 

PLD1 and PLD2 are differentially expressed and modulated in several tissues and cells, and each isoform can be involved in different particular processes [11,19,33,34]. Our previous results constitute the only reports regarding PLDs function and expression in RPE cells. We have demonstrated that in RPE cells exposed to LPS, PKCε activation depends only on PLD1, while PKCα modulation depends on both canonical isoforms [35]. In addition, we have demonstrated that the PLD pathway modulates the autophagic process in RPE cells, PLD2 being involved in basal autophagy and both PLDs modulating autophagy under LPS-induced inflammatory response [23]. Our previous findings postulated for the first time that selective pharmacological PLD1 and PLD2 inhibitors could be promising therapeutic tools to treat or prevent inflammatory events in the retina [20,21,22]. Interestingly, regarding LPS-induced RPE inflammatory response, both isoforms seem to be implicated, as the inhibition of either PLD1 or PLD2 is able to prevent COX-2 induction and PGE2 production [20]. In a similar way, either PLD1i or PLD2i is able to prevent NFκB nuclear translocation, COX-2 and IL-6 induction, and the reduced RPE cell viability induced by HG [21]. These results suggest that the activation of both isoforms is needed, possibly in a concatenated manner or as an enzymatic complex, to mediate the RPE inflammatory response. Here, we demonstrate that HG exposure increases PLD1 and PLD2 mRNA levels in ARPE-19 cells. In agreement with our results, the inhibition or silencing of either PLD1 or PLD2 has been shown to inhibit nicotine- and LPS-induced ERK1/2 and NFκB activation, COX-2 expression, and secretion of tumor necrosis factor α, IL-1 β, and IL-8 in human periodontal ligament cells [36]. In addition, it has been reported that a high glucose environment induces and increases PLD1 expression in pancreatic beta cells [37].

Unlike its role in RPE phagocytosis, the participation of the PLD pathway in leukocyte phagocytosis has been described. Long ago, it was reported that phagocytosis of complement-opsonized particles or activation of Fc gamma receptor (FcgR) activates PLD activity in macrophages and neutrophils [38,39,40,41]. Using an RNA interference strategy, Corrotte and colleagues showed that endogenous PLD1 and PLD2 are necessary for efficient IgG-coated latex bead phagocytosis in RAW264.7 murine macrophages, with both PLD isoforms being required for phagosome formation, although only PLD1 seems to be implicated in later stages of phagocytosis [28]. In a later work, the same group demonstrated the role of the small GTPase RalA in PLD activation elicited after FcgR-mediated phagocytosis in RAW264.7 macrophages [27]. In a similar way, the small GTPase Arf6 was shown to be implicated in the regulation of PLD activity and PA synthesis required for efficient phagocytosis in RAW 264.7 cells [24]. Furthermore, macrophages obtained from *PLD1^−/−^* or *PLD2^−/−^* mice or from wild-type mice whose PLD activity has been pharmacologically inhibited have displayed actin cytoskeleton abnormalities and decreased phagocytic capacity [25,26]. 

Contrary to previous reports in macrophages, our results show that pharmacological inhibition or silencing of PLD1 or PLD2 does not affect RPE phagocytic function or cell viability under physiological conditions, suggesting that PLD-derived PA is not crucial for RPE phagocytosis. In line with this, PA continued to be detected on the phagosome in macrophages obtained from *PLD^null^* mice even when all PLD activity was eliminated [26], demonstrating that PA could be generated by other enzymatic sources, such as DAG kinases. Our findings show that phagocytic function is reduced in ARPE-19 cells exposed to inflammatory injury induced by HG or LPS. In agreement with our results, long-term inflammation induced by LPS reduced viability and POS phagocytosis of primary porcine RPE cultures [42]. Phagocytosis of POS is affected in several pathologies, including diabetes [43,44,45]. Although the molecular mechanisms underlying dysfunctional POS phagocytosis under HG conditions have not been fully elucidated, Puddu and collaborators demonstrated that the expression and activation of MerTK (Mer Tyrosine Kinase), a cell surface receptor that regulates phagocytosis of POS by RPE, is impaired in ARPE-19 cells exposed to HG, leading to defective phagocytosis [46]. 

Furthermore, we have previously shown increased ROS levels in RPE cells exposed to LPS for 24 h or to HG for 72 h [21,23]. In the present work, we demonstrated that ROS levels are increased after a shorter HG exposure (24 h) and that PLD inhibitors are able to prevent ROS generation induced by HG in RPE cells. Interestingly, ROS generation and phagocytic function loss induced by HG are prevented with either PLD1i or PLD2i, suggesting that both isoforms need to be activated to mediate oxidative stress and functionality loss. In a similar way, we previously demonstrated that both canonical PLD isoforms need to be activated to mediate the inflammatory response and cell viability loss in RPE cells exposed to HG [21]. Therefore, our present findings show that targeting either PLD1 or PLD2 pharmacologically prevents phagocytic function loss of ARPE-19 cells exposed to HG. This protective effect on RPE function exerted by PLD1i or PLD2i is probably due to reduced RPE oxidative stress and inflammatory response preventing loss in cell viability and function (Figure 5).

Recently, a novel human RPE cell line named ABC has been characterized [29]. ABC cells spontaneously arose from the eyes of a 19-year-old male donor, and, as mentioned above, they present typical RPE characteristics such as deposition of melanin, formation of microvilli, tight junctions, and honeycomb packing, and expression of specific RPE markers and genes involved in phagocytosis and the visual cycle. In addition, ABC cells present superior phagocytic properties compared to ARPE-19 cells as well as differences in pathways related to senescence, inflammation, and oxidative stress response. Regarding oxidative stress response, ARPE-19 cells are more susceptible to cell death induced by H_2_O_2_ than ABC cells, and the latter undergo contact inhibition without stepping into senescence [29]. Nevertheless, in spite of differences among RPE cell lines, in the present work we demonstrate that PLD1 and PLD2 are expressed in ABC cells and that pharmacological PLD inhibition does not affect ABC cell viability under control conditions, as previously evidenced in ARPE-19 and D407 RPE cell lines [20,21,23]. Furthermore, ABC cells express *PLD1* and *PLD2* to a higher extent than ARPE-19 and hRPE49 cells, and as such they seem to be a good RPE cell line for studying classical PLDs roles. 

Moreover, partial silencing of PLD1 and PLD2 obtained with the combination of both siRNAs was able to reduce mTOR signaling and did not affect ABC phagocytic activity. Although we were not able to achieve full silencing of PLD1 and PLD2, the reduced activation of p70 S6K suggests that partial reduction in PLDs expression is enough to reduce PA-induced mTOR signaling. Similarly, other authors have shown that siRNAs do not always eliminate PLDs expression, while a partial reduction in PLD1 and/or PLD2 expression reduces RAW264.7 macrophages phagocytic capacity [28] or increases autophagy in HEK293 and HeLa cells [47]. 

The mTOR pathway is downregulated in ABC cells compared to ARPE-19 or primary RPE cell culture [29]. Furthermore, the senescence pathway has been shown to be inhibited in ABC cells compared to ARPE-19 cells, suggesting that ABC cells are protected from senescence through the inhibition of the mTOR pathway [29]. Suppression of mTOR is associated with contact inhibition in normal cells, including RPE cells [48,49]. In addition, it has been demonstrated that mTOR pathway activation favors senescence gene programming and expression of the senescence-associated secretory phenotype (SASP), consisting of inflammatory cytokines (IL-1b, IL-6, or IL-8), growth factors, and proteases [50]. In addition, mTOR is crucial for the onset of autophagy, as the active mTORC1 complex prevents the formation of autophagosomes [51]. In line with this, the reduced activation of the mTOR pathway observed in ABC cells in which PLD1 and PLD2 expression was partially reduced suggests that PLD silencing or inhibition could prevent mTOR-induced senescence in RPE cells. Moreover, this result is in agreement with the reduced IL-6 and IL-8 induction observed in ARPE-19 cells exposed to HG and then treated with PLD1i or PLD2i [21] as well as with the enhanced autophagic process observed in RPE cells treated with PLD inhibitors [23]. Regarding mTOR pathway activation in ABC cells, we only observed reduced activation in cells transfected with both PLD1 and PLD2 siRNAs (a condition in which we observed a reduced PLD1 and PLD2 expression), and no difference was observed in cells transfected with only PLD1 siRNA in spite of the reduced PLD1 expression achieved. This fact could be explained by the low PLD1 basal activity compared to PLD2 [13,52]. Further experiments are certainly needed in order to study mTOR modulation by PLD1 and PLD2 in RPE cells exposed to inflammatory conditions.

The morning burst of RPE phagocytosis is coincident with the enzymatic conversion of autophagy protein LC3-I (microtubule-associated protein 1 light chain 3) to its lipidated form (LC3-II), suggesting an interplay of phagocytosis and autophagy within the RPE for POS degradation [53]. In line with this, Dhingra et al. found that the deletion of LC3B resulted in increased phagosome accumulation and that the LC3B isoform is a critical determinant of lipid homeostasis and essential in regulating the inflammatory state of the retina/RPE [54]. On account of the fact that PLD1 and 2 inhibition was able to modulate the autophagic process in RPE cells [23] and that PLD activity is important in macrophage LC3-associated phagocytosis (LAP) [55], further experiments are needed to study the role of the PLD pathway in LAP in this context in greater detail.

## 4. Materials and Methods

### 4.1. Reagents

Sterile dextrose (50% *w/v* in water) was from ROUX- OCEFA S.A. (Buenos Aires, Argentina). Lipopolysaccharide from Klebsiella pneumoniae (LPS, #L4268), Triton X-100 (octyl phenoxy polyethoxyethanol), Phalloidin–Tetramethylrhodamine B isothiocyanate (#P1951), and dimethyl sulfoxide (DMSO) were from Sigma-Aldrich (St. Louis, MO, USA). VU0359595 (PLD1 inhibitor) and VU0285655-1 (PLD2 inhibitor) were from Avanti Polar Lipids, Inc. (Alabaster, AL, USA). DAPI (4′,6-diamidino-2-phenylindole dihydrochloride) was from Life Technologies Corporation (Grand Island, NY, USA). DCDCDHF (5(6)-carboxy-2′7′-dichlorodihydrofluorescein diacetate) was from Molecular Probes (Eugene, OR, USA). All other chemicals were of the highest purity available.

### 4.2. Antibodies

Anti-Rhodopsin antibody (#ab3424) was from Abcam (Cambridge, UK). Rabbit PLD2 (N-term) IgG was from Abgent (San Diego, CA, USA) (#AP14669a). Anti-GAPDH (0411: sc-47724) and m-IgGκ BP-HRP: sc-516102 were from Santa Cruz Biotechnology, Inc. (Dallas, Texas, USA). Anti-PLD1 (#3832), anti-phospho-p70 S6 kinase (Thr389) (#9205) and anti-p70 S6 kinase (#9202) antibodies were from Cell Signaling (Beverly, MA, USA). Polyclonal horse radish peroxidase (HRP)-conjugated sheep anti-mouse IgG (#NA931V) and polyclonal HRP-conjugated donkey anti-rabbit IgG (#NA934V) were purchased from GE Healthcare (Malborough, MA, USA). Alexa Fluor^®^488 goat anti-rabbit (#A11008) was from Life Technologies Corporation (Grand Island, NY, USA). 

### 4.3. Cell Culture and Treatments

The human RPE cell line ARPE-19 from the American Type Culture Collection (ATCC, Manassas, VA, USA) was generously donated by Dr. L. Politi and Dr. N. Rotstein (INIBIBB, Bahía Blanca, Argentina). Cells (in passage 5–15) were propagated in T-25 culture flasks and maintained in Dulbecco’s Modified Eagle’s Medium (DMEM) supplemented with 10% fetal bovine serum (FBS, Natocor, Argentina) and antibiotic-antimycotic (Anti-Anti 100X, Gibco by Life Technologies, Grand Island, NY, USA) at 37 °C under 5% CO2. Except for microscopy experiments, for the remaining assays all cells were grown to 100% confluence. ARPE-19 cells were serum-starved for 1 h and subsequently exposed to normal glucose concentration (Control condition or NG, 5.5 mM) or to high glucose concentrations (HG, 33 mM) for 24, 48, or 72 h. To maintain glucose concentrations, the medium was replaced every 24 h. In addition, after serum starving, ARPE-19 cells were treated with LPS (10 μg/mL or 25 μg/mL) for 24 or 48 h in serum-free DMEM or with the same volume of sterile ultra-pure water (control condition). LPS stock (4 mg/mL) was prepared in sterile ultra-pure water. To study the role of the PLD pathway, cells were pre-incubated with different concentrations (0.5 or 5 μM) of VU0359595 (PLD1i) to inhibit PLD1 activity or with different concentrations (0.5 or 5 μM) of VU0285655-1 (PLD2i) to inhibit PLD2 for 30 min at 37 °C prior to cell exposure to HG. PLD inhibitors were present during HG treatment, and DMSO (vehicle of PLD inhibitors) was added to all conditions to achieve a final concentration of 0.0025% or 0.025% depending on the concentration of PLDi used. When the effect of both PLDi concentrations (0.5 or 5 μM) was studied in the same experiment, DMSO was added to all conditions in the maximal vehicle concentration (0.025%).

ABC cells are a novel human RPE cell line derived from a primary culture obtained from the ocular globes of a 19-year-old male donor [29]. ABC cells were cultured in T-75 flasks in Minimum Essential Media (MEM, Millipore Sigma, Burlington, MA) containing 10% fetal calf serum, 5% newborn calf serum, non-essential amino acids, 4 mM glutamine, amphotericin B (0.5 g/mL), and gentamicin (10 μg/mL) at 37 °C under 5% CO2. Cells from passage 20–25 were used in all experiments. To inhibit classical PLDs in ABC cells, 5 μM of VU0359595 (PLD1i) and 5 μM of VU0285655-1 (PLD2i) were used. DMSO (vehicle of PLD inhibitors) was added to all conditions to achieve a final concentration of 0.025%. 

### 4.4. Measurement of Reactive Oxygen Species (ROS) Production

ROS production was measured using a DCDCDHF probe (Molecular Probes, Eugene, OR), as previously described [21]. Briefly, 60,000 ARPE-19 cells were seeded onto 12 mm coverslips and exposed to NG, HG (33 mM), HG plus PLD1i (0.5 μM), or HG plus PLD2i (0.5 μM) for 24 h. After the experimental treatment, cell culture medium was removed and replaced by medium containing 10 μM DCDCDHF and cells were incubated for 30 min at 37 °C. Cells were subsequently washed three times with PBS and coverslips were mounted for examination with a Nikon Eclipse TE2000-S microscope coupled to a Nikon DS-Qi2 camera (1608 × 1608 pixels) and a 63× Plan Apo (1.4 N.A.) oil-immersion objective. Fluorescence intensity values were determined using ImageJ 1.46 software.

### 4.5. Non-Specific Phagocytosis Assays

Non-specific phagocytosis was measured using pHrodo™ green bioparticles^®^ conjugates (Molecular Probes, Eugene, OR, USA, #P35366) following the manufacturer’s instruction. pHrodo™ BioParticles^®^ conjugates are novel no-wash fluorogenic reagents developed for quantitative measurements of phagocytosis. Briefly, ARPE-19 cells (1.5 × 10^4^ cells/well) were seeded in 96-well plates; after reaching confluence, cells were exposed to the experimental treatments described above. Then, medium was removed and 100 μL/well of bioparticles resuspended in serum free DMEM (1 mg/mL) were added. After 4 h incubation, medium was removed, cells were washed three times with PBS, and fluorescence was measured using a fluorescence plate reader Fluoroskan Ascent FL (Thermo Fisher Scientific, Waltham, MA, USA, λex 485 nm-λem 538 nm). Results are expressed as arbitrary units (AU). Control of autofluorescence level was performed with cells non-incubated with bioparticles. Representative images were taken from the plate using a Nikon Eclipse TE2000-S microscope coupled to a Nikon DS-Qi2 camera (1608 × 1608 pixels) and a 20× objective.

### 4.6. POS Phagocytosis Assays

ARPE-19 and ABC cells were exposed to experimental treatments and then incubated with unlabeled POS (rod bovine outer segments, InVision BioResources, Seattle WA, # 98740) suspension at a 1:10 (cell:POS) ratio in DMEM or MEM for 16 h [29]. To stop the reaction, cells were washed three times with PBS-CM buffer (PBS supplemented with 1 mM MgCl_2_, 0.2 mM CaCl_2_). To detect internalized POS only, cells were washed one time with PBS, then incubated with PBS-EDTA for 5–10 min. Samples designated for total POS detection remained in PBS-CM. PBS-EDTA was then removed and cells were washed three times with PBS-CM. Phagocytosis was assessed by immunocytochemistry or by western blotting (WB) assays using an anti-rhodopsin antibody [29,30]. For WB, PBS-CM was removed from all wells and cells were lysed with RIPA buffer (10 mM Tris-HCl (pH 7.4), 15 mM NaCl, 1% Triton X-100, 5 mM NaF, 1 mM Na_2_VO_4_) freshly supplemented with protease inhibitor cocktail. For POS phagocytosis measurement by immunocytochemistry. ARPE-19 cells were grown onto coverslips and, after POS phagocytosis assay, cells were fixed with 2% paraformaldehyde in PBS at room temperature for 15 min, permeabilized with 0.1% Triton X-100 in PBS for 15 min, and blocked with 2% BSA for 15 min. Cells were then incubated with the primary anti-Rhodopsin antibody (1:300 in blocking solution) for 1 h and Alexa Fluor^®^488-conjugated secondary antibody (1:500 in blocking solution) for 1 h. Phalloidin was used to visualize F-actin, and nuclei were stained with DAPI. Cells were imaged using a Zeiss LSM 900 confocal microscope confocal and a 63x Plan Apo (1.4 N.A.) oil-immersion objective or a Nikon Eclipse TE2000-S microscope and a 63× Plan Apo (1.4 N.A.) oil-immersion objective. We counted 300 cells in a blinded manner per condition in order to determine the percentage of cells with phagocyted POS. 

### 4.7. Western Blot (WB) Assays

After experimental treatments, the medium was removed from confluent cultures, then cells were washed three times with PBS and scraped off with 80 μL ice-cold RIPA lyses buffer (10 mM Tris-HCl (pH 7.4), 15 mM NaCl, 1% Triton X-100, 5 mM NaF, 1 mM Na_2_VO_4_, and complete protease inhibitor cocktail). Protein content of total cell lysates was determined by the Bradford method [56] (Bio-Rad Life Science group, #500-0006) and samples were denatured with Laemmli sample buffer at 95 °C for 5 min. Equivalent amounts of proteins (30 μg) were separated by sodium dodecylsulfate polyacrylamide gel electrophoresis (SDS-PAGE) and transferred to polyvinylidene fluoride (PVDF) membranes (Millipore, Bedford, MA, USA). WB assays were performed as previously described [20,35]. Briefly, after being blocked with 10% BSA in TTBS buffer (20 mM Tris–HCl (pH 7.4), 100 mM NaCl and 0.1% (w/v) Tween 20) at room temperature for 2 h, membranes were subsequently incubated overnight at 4 °C with primary antibodies, washed three times with TTBS, and exposed to the appropriate HRP-conjugated secondary antibody for 2 h at room temperature. Immunoreactive bands were detected by enhanced chemiluminescence (Pierce^®^ ECL Western Blotting Substrate, #32209, Thermo Fisher Scientific, Waltham, MA, USA) using UltraCruz^®^ Autoradiography Film, Santa Cruz Biotechnology, Inc. (Santa Cruz, CA, USA) or visualized using a Bio-Rad Chemidoc imaging system (Bio-Rad Life Science group). Densitometry values of the immunoreactive bands were determined using ImageJ 1.46 software. The molecular weight of bands was determined using the spectra multicolor broad range protein ladder (#26634, Thermo Fisher Scientific, Waltham, MA, USA) or Bio-Rad Precision Plus Protein Dual Color Standards (#1610374. Bio-Rad Life Science group) and Biotinylated Protein Ladder Detection Pack (#7727, Cell Signaling Technology, Inc) in the case of ABC cell Western Blots.

### 4.8. Real-Time Quantitative PCR (qPCR) Assays

RT-qPCR assays were performed as previously described [21]. Total RNA was isolated from treated ARPE-19 cells (confluent 35 mm dishes) using 500 µL TRIzol™ reagent (Invitrogen, Carlsbad, USA) following the manufacturer’s instructions. The RNA was resuspended in RNase-free water and its concentration and purity were assessed from the A260:A280 absorbance ratio in a PicoDrop Spectrophotometer. Total RNA (2 µg) was used to synthesize cDNA by reverse transcription (RT) in a final volume of 25 µL containing 1 μg Random Primers hexamers (Biodynamics, Buenos Aires, Argentina), 1x M-MLV RT Reaction Buffer, 0.5 mM of each dNTP, 25 UI RNase inhibitor (Promega, Madison, WI, USA), and 200 UI M-MLV Reverse Transcriptase (Promega, Madison, WI, USA). The cDNA resulting from the RT was amplified by real-time quantitative PCR (qPCR).

The qPCR assays were performed in a final volume of 10 μL using 0.2 μM of each primer and KAPA SYBR^®^ FAST qPCR Kit Master Mix (Kapa Biosystems, Boston, MA, USA). Primer sequences for the specific amplification of PLD1, PLD2, and GADPH are listed in Table 1 and were purchased from Invitrogen Life Technologies. Gene expression levels were determined using a Rotor-Gene 6000 Series Software machine (Corbett Research, Australia) with the following conditions: 35 cycles of denaturation at 94 °C for 20 s, annealing and extension at 58 °C for 30 s, and a final extension step at 72 °C for 30 s. Ct values of PLD1 and PLD2 mRNA obtained from three different experiments were normalized using GAPDH as a reference gene, and relative quantification was performed applying the 2^−ΔΔCt^ method [57].

### 4.9. PLD Silencing

Transient siRNA transfections were carried out. ABC cells were plated in 12-well plates in complete medium. The following day, transfections with siRNA (100 nM) using Lipofectamine 2000 (Thermo Fisher Scientific, Waltham, MA, USA) were performed in Opti-MEM reduced serum medium (Thermo Fisher Scientific, Waltham, MA, USA). After 48 h, cells were shifted to complete medium. The following siRNAs were used in this study: PLD1 Human Silencer^®^ Select Assay ID s10637, PLD2 Human Silencer^®^ Select Assay ID s10641, and Ambion Negative control (all three from Thermo Fisher Scientific, Waltham, MA, USA) and AllStars Neg. siRNA AF 555 (Qiagen, Hilden, Germany). The latter was used as a transfection control in all the experimental samples.

### 4.10. RNAseq Analysis and Bioinformatic Analyses of RNAseq

RNAseq analysis was performed as previously described [29]. For RNAseq, ABC cells, ARPE-19 cells, and hRPE49 cells were seeded in 6-well plates (500,000 cells/well) and grown for at least 72 h. Trizol was used to extract total RNA, and its quality and integrity were verified using NanodropOne (Thermo Fisher Scientific) and Agilent 2100 Bioanalyzer RNA nanochips. Library preparation was carried out using TrueSeq RNA library prep kit v2—Set A (catalog #RS-122-2001) (Illumina); libraries were denatured, pooled, and normalized according to Illumina’s sample preparation guide. RNASeq was carried out on an Illumina NextSeq 500 sequencing system using NextSeq 500 High Output v2 kit (150 cycles) (catalog# FC-404-2002).

The RNA reads were mapped to the human genome and processed through the bioinformatics pipeline in QIAGEN CLC Genomics Workbench (Qiagen, Redwood City, CA, United States). Then, the results were analyzed using R and the heat map of PLD1 and PLD2 gene expression was generated with Heatmap Generator. 

### 4.11. Flow Cytometry

Confluent ABC cells were treated for 1 h with vehicle (0.025% DMSO), PLD1i (5 μM), PLD2i (5 μM), or their combination (PLD1i and PLD2i, 5 μM); 16 h later, cells were labeled with propidium iodide (PI), which is a membrane impermeant dye that is generally excluded from viable cells, and analyzed immediately on a Gallios Flow Cytometer (Beckman Coulter, Brea, CA, USA). After excluding cell debris and doublets by proper gating, cells were gated between PI positive (not viable) and PI negative (viable). Ten thousand events were collected per treated sample. Data were analyzed using Kaluza Analysis Software (Beckman Coulter, Brea, CA, USA). 

### 4.12. Statistical Analysis

Statistical analysis was performed using ANOVA followed by Bonferroni’s test to compare means; *p*-values lower than 0.05 were considered statistically significant. Data represent the mean value ± SEM of at least three independent experiments. The WBs and microscopy images shown are representative of three independent experiments.

## 5. Conclusions

Our previous findings together with the results presented herein demonstrate that pharmacological targeting of classical PLDs can prevent inflammatory response, oxidative stress, and the loss of phagocytic function and cell viability of RPE cells exposed to high glucose levels without affecting RPE function or viability under non-inflammatory conditions. Due to the relevance of RPE in visual function and in PR survival, we are encouraged to move forward to further study the role of the PLD pathway in PR cells and to test the effect of PLD pharmacological inhibitors on visual function in vivo.

## Figures and Tables

**Figure 1 ijms-23-11823-f001:**
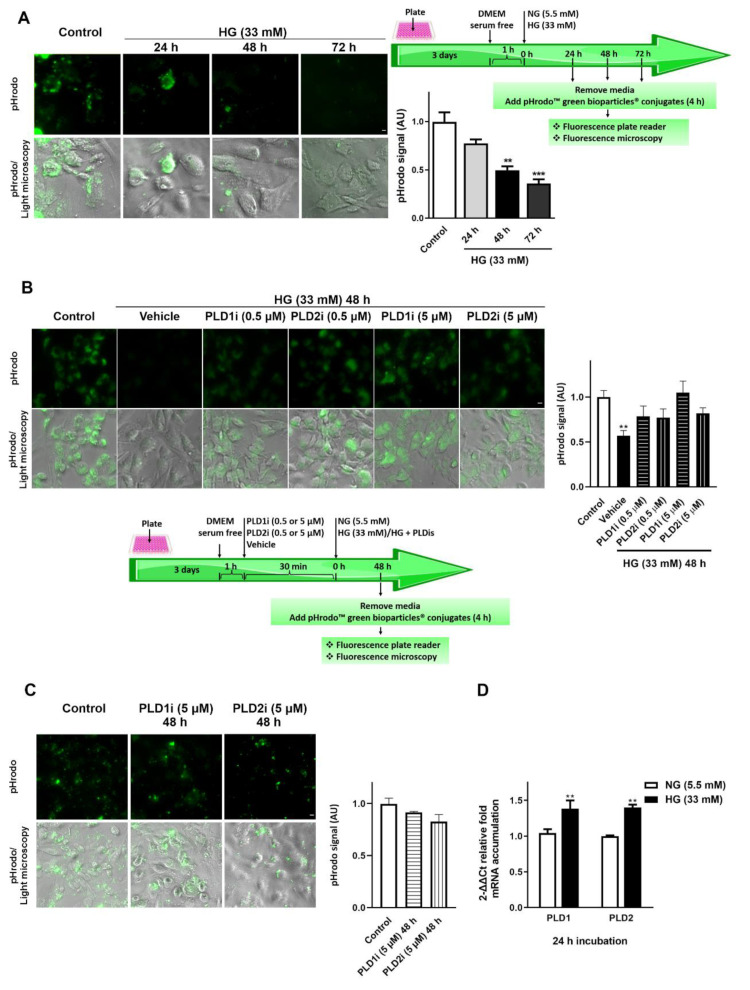
Non-specific phagocytosis in ARPE-19 cells exposed to inflammatory injury. (**A**) ARPE-19 cells were seeded in 96-well plates and exposed to high glucose concentration (HG, 33 mM) for 24, 48, or 72 h. (**B**) ARPE-19 cells were exposed to vehicle 0.025% DMSO (control condition), PLD1i (0.5 or 5 μM) or PLD2i (0.5 or 5 μM) for 30 min prior to cell exposure to 33 mM HG or to NG (control condition). PLD inhibitors were present during HG treatment. (**C**) ARPE-19 cells were exposed to vehicle 0.025% DMSO (control condition), PLD1i (5 μM), or PLD2i (5 μM) for 48 h in media containing normal glucose concentration (Control condition or NG, 5.5 mM). For A-C, non-specific phagocytosis was measured using pHrodo™ green bioparticles^®^ conjugates as described in the Materials and Methods section. Representative fluorescence images (Scale bar = 10 µm) are shown; bar graphs show pHrodo fluorescence intensity expressed as arbitrary units (AU) with respect to control conditions. Significant differences with respect to each control condition are indicated with asterisks (*** *p* < 0.001; ** *p* < 0.01). (**D**) ARPE-19 cells were exposed to HG (33 mM) or NG (5.5 mM) for 24 h. qPCR assays for the quantification of PLD1 and PLD2 mRNA levels were performed as described in the Materials and Methods section. Results are expressed as 2^-ΔΔCt^ relative fold mRNA accumulation using GAPDH as internal reference gene. Significant differences with respect to NG are indicated with asterisks (** *p* < 0.01).

**Figure 2 ijms-23-11823-f002:**
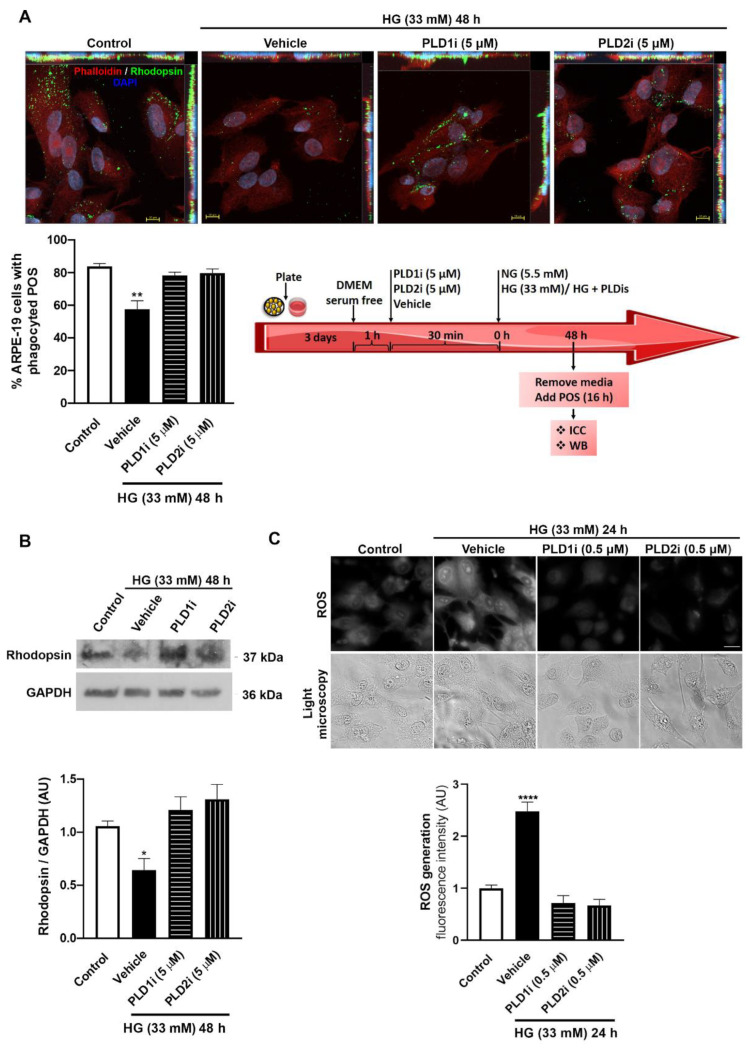
Effect of PLD1i and PLD2i on photoreceptor outer segments (POS) phagocytosis by ARPE-19 cells exposed to HG. For A and B, ARPE-19 cells were seeded onto coverslips and pre-incubated with vehicle (0.025% DMSO), PLD1i (5 μM), or PLD2i (5 μM) for 30 min, then exposed to control condition (NG plus vehicle, 0.025% DMSO), HG (33 mM) plus vehicle, HG plus PLD1i (5 μM), or HG plus PLD2i (5 μM) for 48 h. Then, medium was removed and cells were incubated with unlabeled POS suspension in DMEM at a 1:10 (cell:POS) ratio for 16 h and internalized POS were detected by immunocytochemistry assays as described in the Materials and Methods section. (**A**) Representative confocal microscopy images (z-stacks maximum intensity projection) show rhodopsin staining (green), DAPI (blue) and phalloidin staining (red). Orthogonal view from different planes (x/y, x/z or y/z) of the confocal microscope images are shown (Scale bar = 10 μm). Bar graph shows the % of ARPE-19 cells with internalized POS. Asterisks (*) indicate significant differences with respect to control condition (** *p* < 0.01). (**B**) Internalized POS were measured by WB to detect rhodopsin and GAPDH. Molecular weights are indicated with numbers to the right. Bar graph shows the densitometry values of rhodopsin/GAPDH expressed as arbitrary units (AU) with respect to control conditions (NG). Significant differences with respect to NG are indicated with asterisks (* *p* < 0.05). The whole membranes are depicted in Appendix A. (**C**) Effect of PLD1i and PLD2i on reactive oxygen species (ROS) generation in ARPE-19 cells exposed to HG. ARPE-19 cells were seeded onto coverslips and pre-incubated with vehicle (0.0025% DMSO), PLD1i (0.5 μM), or PLD2i (0.5 μM) for 30 min, then exposed to control condition (NG plus vehicle, 0.0025% DMSO), HG (33 mM) plus vehicle, HG plus PLD1i (0.5 μM), or HG plus PLD2i (0.5 μM) for 24 h. ROS production was measured using DCDCDHF. Representative fluorescence microscopy images (first row) and light microscopy images (second row) are shown (Scale bar = 20 µm). Bar graph shows fluorescence intensity expressed as arbitrary units (AU) with respect to control conditions. Significant differences with respect to NG are indicated with asterisks (**** *p* < 0.0001).

**Figure 3 ijms-23-11823-f003:**
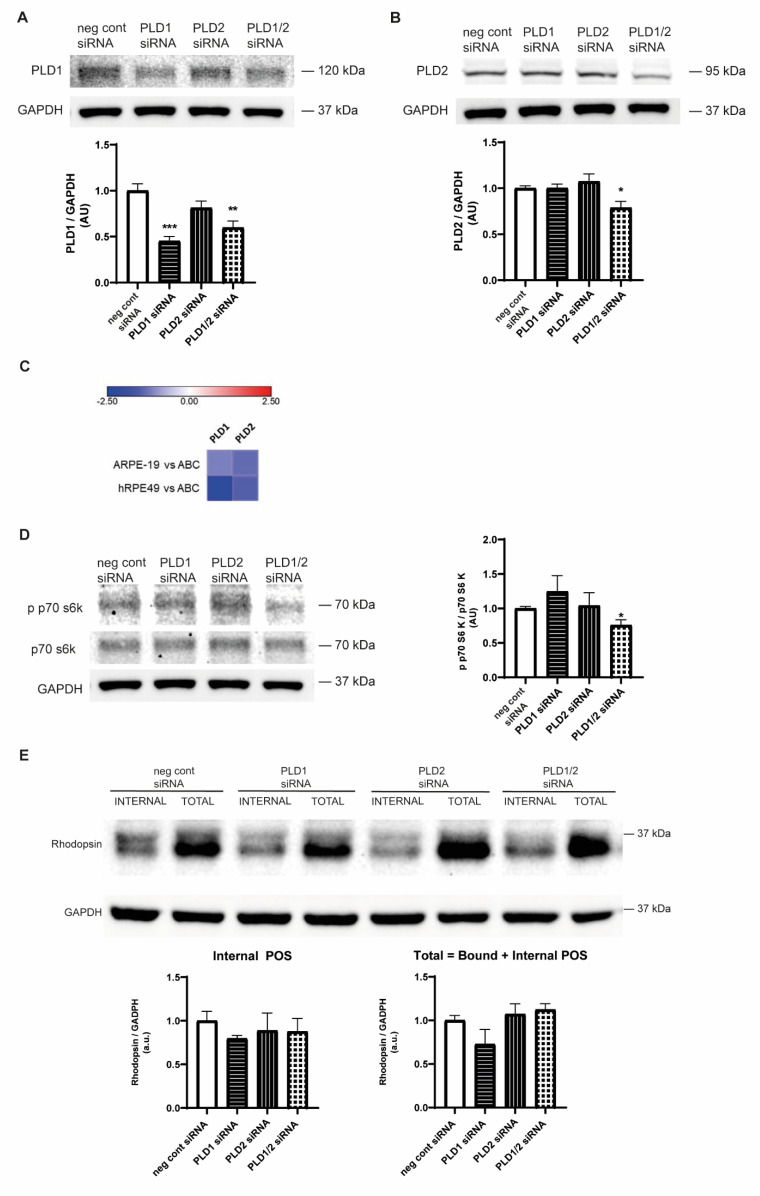
PLD1 and PLD2 expression and silencing in ABC cells. Confluent ABC cell cultures were transiently transfected for 48 h with PLD1 siRNA, PLD2 siRNA, or PLD1 plus PLD2 siRNAs, as described in the Materials and Methods section. WB assays were performed to detect PLD1 (**A**), PLD2 (**B**), and GAPDH. For A and B, bar graphs show the densitometry values of PLD1/GAPDH and PLD2/GAPDH expressed as arbitrary units (AU) with respect to control conditions (neg control siRNA). (**C**) Heatmap from RNAseq analysis of ARPE-19 cells and hRPE49 cells compared to ABC showing PLD1 and PLD2 expression. Analysis was performed using ABC cells as control (fold change). Red means the pathway is upregulated when compared to ABC, while blue means it is downregulated. Data are detailed in Appendix A. (**D**) Effect of PLD silencing on mTOR pathway activation. WB was performed to detect activation (phosphorylation) of p70 S6K (p p70 S6K), total p70 S6K, and GAPDH. The bar graph shows the densitometry values of p p70 S6K/p70 S6K expressed as arbitrary units (AU) with respect to control conditions (neg control siRNA). (**E**) Effect of PLD silencing on POS phagocytosis of ABC cells. After siRNA transfection, the medium was removed and ABC cells were incubated with unlabeled POS suspension in MEM at a 1:10 (cell:POS) ratio for 16 h as described in the Materials and Methods section. Total POS (bound and internalized) and internal POS were determined by WB. Bar graphs show the densitometry values of rhodopsin/GAPDH expressed as arbitrary units (AU) with respect to control conditions (neg control siRNA). For (**A**,**B**,**D**,**E**), numbers to the right indicate molecular weights, asterisks (*) indicate significant differences with respect to control condition (*** *p* < 0.001; ** *p* < 0.01; * *p* < 0.05), and whole membranes are depicted in Appendix A.

**Figure 4 ijms-23-11823-f004:**
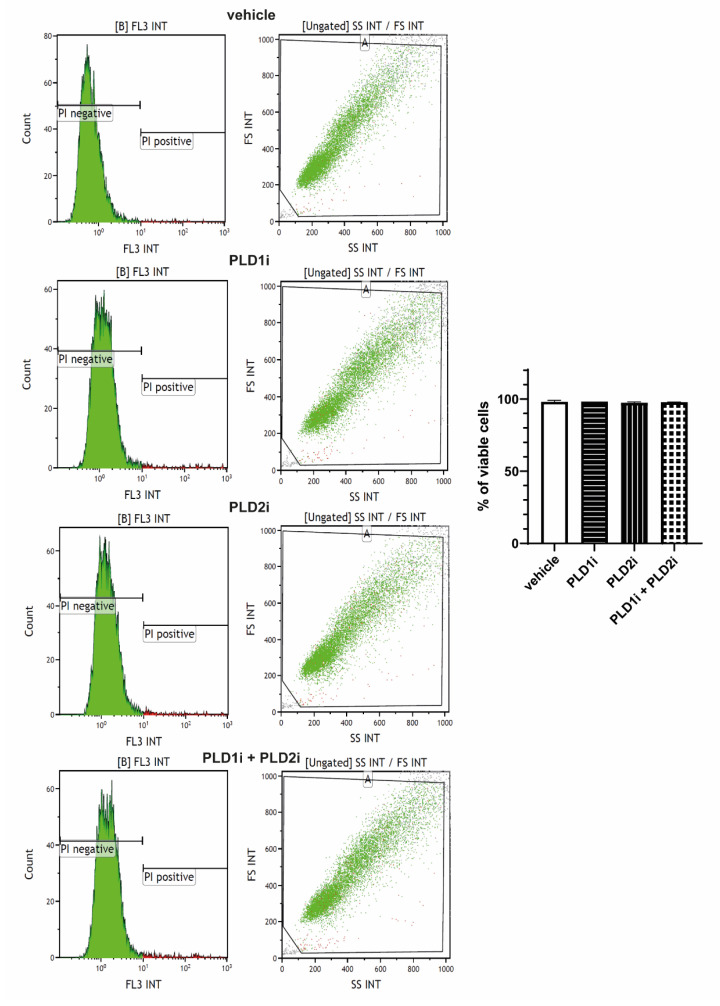
Effect of PLDi on viability of ABC cells. Confluent ABC cells were treated for 1 h with vehicle (0.025% DMSO), PLD1i (5 μM), PLD2i (5 μM), or their combination (PLD1i and PLD2i, 5 μM); 16 h later, cells were labeled with propidium iodide (PI) (see Materials and Methods) for flow cytometry analysis. After excluding cell debris and oublets by proper gating, cells were gated between PI positive (not viable) and PI negative (viable). Ten thousand events were collected per treated sample. Data were analyzed using Kaluza Analysis Software (Beckman Coulter, Brea, CA, USA).

**Figure 5 ijms-23-11823-f005:**
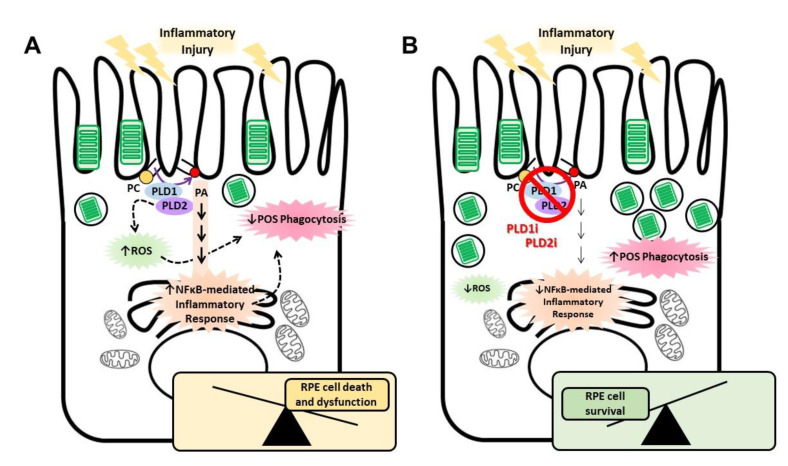
Schematic view of PLD-mediated events elicited in RPE cells exposed to inflammatory injury. (**A**) The PLD pathway is activated in RPE cells exposed to inflammatory injury to mediate reactive oxygen species (ROS) generation and nuclear factor kappa B (NFκB)-mediated inflammatory response, leading to a reduced RPE cell viability and phagocytic function. (**B**) Pharmacological inhibition of PLD1 and PLD2 is able to prevent ROS generation and the RPE inflammatory response, resulting in a normal phagocytic function. Dashed arrows indicate possible but not fully elucidated mechanisms.

**Table 1 ijms-23-11823-t001:** Primer sequences for the specific amplification of human *PLD1, PLD2*, and *GAPDH*.

Gene Name	Gene Symbol	Sequence Accession Number	Primer Sequences 5′–3′	Amplicon (pb)
Phospholipase D1	*HUMAN_PLD1*	NM_002662.4	TGGAAGAGGCAAATGAAGAGACAACGATTTCCCTCAACCAC	91
Phospholipase D2	*HUMAN_PLD2*	NM_002663.4	CCAAGTTTGTTATCGCTGGTCCAAATGAGATGGCACCTGTC	92
Glyceraldhehyde-3-phosphate dehydrogenase	*HUMAN_GAPDH*	NM_001289746.1	CACTGAATCTCCCCTCCTCACATGATGGTACATGACAAGGTGCG	87

## Data Availability

The original contributions presented in this study are included in the article/Appendix A; further inquiries can be directed to the corresponding author.

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
