# Peer review of "Targeting Phospholipase D Pharmacologically Prevents Phagocytic Function Loss of Retinal Pigment Epithelium Cells Exposed to High Glucose Levels"

_ijms, 2022, doi:10.3390/ijms231911823_

Round 1

Reviewer 1 Report

The manuscript of Bermudez V and al. describes the effects of pharmacological PLD inhibition on phagocytosis in RPE cells after high glucose treatment. 

The authors use PLD inhibitors and siRNAs to inhibit PLD activity in two RPE cells: ARPE-19 and ABC.

Although the reported experiments are well addressed to this focus, they are incomplete.

1) Although the authors show the effects of PLD inhibitors on phagocytosis of ARPE-19 cells by immunofluorescence (pHrodo particles, Fig. 1 and Rhodopsin, Fig. 2A) and western blotting by rhodopsin (Fig.2B), however, they do not show the activity of PLDs. They have to show the immunoblotting of p70S6K phosphorylation as a marker for mTORC1 activity as performed in ABC cells. In addition, does the PLD activity decrease and the PLD downregulation by siRNA reduce the total Rhodopsin amount?

2) In figure 2, it is not clear whether the  PLD inhibitors are co-incubated or pre-incubated with HG in ARPE-19 cells. In the text, the authors reported that they are co-incubated while the scheme in 2A shows that they are pre-incubated. This reviewer suggests performing the experiment pre-incubating with PLD inhibitors

3) the PLD siRNA combination appears to decrease the PLD1 and 2 in ABC cells. Did the authors test them in ARPE19 cells? did the authors analyze the phagocytosis in ARPE-19 siRNA cells?

4) Fig.4D show a 25% decrease of p-p70S6K1 after PLD1/2 down-regulation and no changes in Rhodopsin levels. Why is not the Rhodopsin level affected?

5) The authors should use the PLD chemical inhibitors (PLD1i and PLD2i) in ABC cells to test the mTORC1 activity and Rhodopsin level as previously performed by PLD siRNAs

Author Response

Reviewer 1

Thank you very much for your observations and comments, we really appreciate them. Please find below a point-by-point response to your comments.

Reply to point 1: We reported PLD activation in HG-exposed ARPE-19 cells in a previous work from our laboratory (Tenconi et al, 2019. Reference 21 in the present paper). In this previous publication we characterized HG-induced injury in ARPE-19 cells and we demonstrated that PLD activity was increased by 80% in cells exposed to HG for 4 h. Furthermore, by using PLD1 and PLD2 pharmacological inhibitors we demonstrated that both canonical PLDs are activated after 4 h HG treatment. This point has been clarified in the revised manuscript (see lines 122-126).

In the present work we demonstrated the expression of PLD1 and PLD2 in ABC cells, a novel human RPE cell line recently described (Asatryan et al, 2022. Reference 29 in the present paper), and also PLD1 and PLD2 gene expression in hRPE49 cells. Certainly, further experiments are needed to characterize the effect of HG as well as to study the activation of PLDs in this novel RPE cell line. In the present work we show that p70S6K phosphorylation, a downstream mTORC1 effector, is reduced in cells transfected with PLD1 and PLD2 siRNA (Figure 3D) suggesting a reduce PLD-derived PA generation. However, in ABC cells transfected with PLD1 and PLD2 siRNA no significant differences in POS phagocytosis (measured by WB with an anti-rhodopsin antibody, Figure 3E) were observed. This result suggests that PLD activity is not necessary for POS phagocytosis under non-inflammatory basal conditions and is in agreement with results obtained with PLD inhibitors in ARPE-19 cells exposed to control conditions (Figure 1C). This point has been clarified in the revised manuscript (see lines 207-208 and 216-219).

Reply to point 2: Thank you for this observation. As mentioned in Materials and Methods section and as shown in figures’ timelines, in all experiments performed with PLD inhibitors cells were pre-incubated with PLD1i or PLD2i for 30 min before HG exposure and then inhibitors were also present (co-incubated) during HG treatment in HG-PLD1i and HG-PLD2i conditions. This has been clarified in the result section and figure legends of the revised manuscript (please see lines 127-129, 132, 152, 163-164, 179-180).

Reply to point 3: We have not been able to obtain significant silencing of PLD1 and PLD2 in ARPE-19 cells using siRNA yet. Nevertheless, we were interested in testing the effect of PLD inhibitors in the phagocytic function of ARPE-19 cells in order to evaluate the potential use of these pharmacological inhibitors as therapeutic agents for retinal inflammatory conditions.

Reply to point 4: Since RPE cells do not express rhodopsin, the detection of this protein in RPE cells is a reliable indicator of POS phagocytosis. In spite of the reduced p70 S6K activation, in ABC cells transfected with PLD1 and PLD2 siRNA no significant differences in POS phagocytosis (measured by WB using an anti-rhodopsin antibody, Figure 3E) were observed. This result suggests that PLD activity is not necessary for POS phagocytosis under non-inflammatory basal conditions and is in agreement with results obtained with PLD inhibitors in ARPE-19 cells exposed to control conditions (Figure 1C). This point has been clarified in the revised manuscript (see lines 149-150, 216-219).

Reply to point 5: Thank you very much for your suggestions. In view of the modulation of the mTOR pathway by PLDs and the fact that it has been demonstrated that mTOR pathway activation favors senescence, is crucial for the onset of autophagy and that its suppression is associated with contact inhibition in normal cells, we are planning to study in more detail the relationship between classical PLDs and mTOR pathway in RPE cells exposed to inflammatory injury in future works. This has been discussed in the present manuscript (see lines 369-381).

Reviewer 2 Report

1/ On Line 94, authors mentioned that pHrodo Red E-coli BioParticles was used to study the non-specific phagocytosis. However, in the method part, authors stated they were using pHrodo green bioparticles conjugates for studying non-specific phagocytosis on Line 442. Could authors clarify which one was used in this study?

2/ In Figure 1B and 1C, which concentration of DMSO (0.0025% or 0.025%) was used in the vehicle control? Authors should use both concentrations as vehicle controls in Figure 1B as both 0.5 mM and 5 mM of inhibitors were used.

3/ On Line 100, authors mentioned they treated the ARPE-19 cells with LPS (25 mg/ml) for 48 hours and showed in Supplementary Figure 1. Could authors also show the treatment at 24 hours, which readers can compare the effect between 10 mg/ml LPS at both 24 hours and 48 hours then.

4/ In Figure 1D, both PLD1 and PLD2 mRNA expressions were detected after 24 hours NG or HG incubation in ARPE-19 cells. However, authors treated the ARPE-19 cells with HG for 48 hours in all the tests (Figure 1B and 1C). Could authors show the mRNA expressions of both PLD1 and PLD2 after 48 hours of HG treatment? This is important to perform the tests with the same time point for better comparison or a clearer picture.

5/ Authors pre-treated the cells with inhibitors for 30 mins prior to cell exposure to HG or NG in Figure 1B. Did authors check the PLD1 or/and PLD2 expressions were suppressed after treatment of inhibitors?

6/ In Figure 3B, siPLD2 could not reduce the expression of PLD2. Is the amount of siRNA not enough or could authors explain why? Moreover, the both siRNAs of PLD1 and PLD2 could suppress the PLD2 expression instead of PLD2 siRNA, could authors also discuss more about this observation?

7/ Similar to #6, in Figure 3D, the pp70 s6k increased when cells treated with either siPLD1 or siPLD2, but it was reduced when cells treated with both siRNAs together. Could authors discuss the possible reasons behind?

8/ Why authors used ABC cells in this study? Could authors discuss more about this or compare which one is better (ARPE-19 or ABC cells)?

Author Response

Reviewer 2

Thank you very much for your observations and comments, we really appreciate them. Please find below a point-by-point response to your comments.

Reply to point 1: Thank you very much for noticing this typing error. In this paper non-specific phagocytosis has been assayed using only pHrodo green bioparticles. This mistake has been amended in the revised manuscript.

Reply to point 2: In figures 1B and 1C 0.025% DMSO was added to the control condition, as stated in the figure legend. When the effect of both PLD inhibitors concentrations (0.5 and 5 μM) was studied in the same experiment DMSO was added to all conditions in order to achieve the maximal vehicle concentration (0.025%). We have previously determined that this DMSO concentration does not affect ARPE-19 and D407 RPE cells viability. This point has been clarified in Materials and Methods section of the revised manuscript (see lines 433-434).

Reply to point 3: Data showing non-specific phagocytosis in ARPE-19 cells exposed to 25 μg/ml LPS for 24 h have been included in supplementary figure 1 of the revised version.

Reply to point 4: In our previous work we demonstrated that PLD activity is increased by 80% in ARPE-19 cells exposed to HG (33 mM) for 4 h (Tenconi et al, 2019. Reference 21 in the present paper). In the same work we also demonstrated that HG exposure for 24 h increases IL-6, IL-8 and COX-2 mRNA levels and that this increment in pro-inflammatory mediators’ expression was prevented by PLD inhibitors, that is why we decided to study the effect of HG on PLD1 and PLD2 expression after 24 h incubation. However, the effect of PLD1i and PLD2i on RPE phagocytosis after 48 h exposure to HG since phagocytosis was not significantly affected by 24 h HG exposure (Figure 1A). This point has been clarified in the revised manuscript (see lines 122-126, 136-139).

Reply to point 5: We have previously demonstrated that PLD1i and PLD2i reduce HG-induced PLD activity in ARPE-19 cells (Tenconi et al, 2019. Reference 21 in the present paper). We did not evaluate the effect of these inhibitors on PLD1 and PLD2 expression. However, pharmacological inhibitors do not always induce a suppression in protein expression and both PLD inhibitors neither affect RPE cell viability nor functionality under non-inflammatory conditions.

Reply to point 6: Effectively, we have not been able to obtain a significant silencing of PLD2 in ABC cells using only PLD2 siRNA yet. We tried different siRNA concentrations, different time incubations and different amounts of lipofectamine. Moreover, as already mentioned in the responses to reviewer 1, in ARPE-19 cells we have not been able to obtain significant silencing of PLD1 or PLD2 using siRNA yet. But we could partially silence PLD1 and PLD2 using both PLD1 and PLD1 siRNAs at the same time in ABC cells, similar to what other authors could achieve in other cell types as already point out in the discussion of the main text (lines 356-359). PLD2 activity is constitutively higher than that of PLD1 in mammalian cells, a factor that could explain why it is more difficult to silence than PLD1. Other approaches could be tested to improve PLDs silencing in RPE cell lines in future works.

Reply to point 7: Regarding mTOR pathway activation, in cells transfected with either PLD1 or PLD2 siRNA we did not observe any significant differences in p70S6K phosphorylation with respect to the control condition. On the contrary, we observed a reduced p70S6K phosphorylation in ABC cells transfected with both PLD1 and PLD2 siRNAs (condition in which we observed a reduced PLD1 and PLD2 expression, 40% and 21% respectively) and no difference was observed in cells transfected with only PLD1 siRNA, in spite of the reduced PLD1 expression observed (55%). This fact could be explained by the low PLD1 basal activity compared to PLD2 and was discussed in the manuscript (lines 374-381). Certainly, further experiments are needed to study mTOR modulation by the PLD pathway in RPE cells.

Reply to point 8: Although the activity and expression of PLD1 and PLD2 have been studied in several physiological and pathological processes, very little is known about the expression of these signaling enzymes in ocular tissues and retinal cells as well as their role in visual function and ocular response to stress. Our previous findings constitute the only reports regarding classical PLDs in RPE cells. We have previously reported PLD1 and PLD2 expression in ARPE-19 and D407 RPE cells and discussed their role in the inflammatory response and the autophagic process. Here we also demonstrate the expression of these signaling enzymes in ABC and hRPE49 cells. Furthermore, we also show that PLD1i and PLD2i do not affect ABC cells viability either, in agreement with our previous results obtained in ARPE-19 and D407 cells. This point has been mentioned in the manuscript (see lines 242-248)

ABC cells have been recently described and compared to ARPE-19 and differences in pathways related to senescence, inflammation, oxidative stress response, as well as markers of native RPE were described. In regard to the oxidative stress response, ARPE-19 are more susceptible to cell death induced by H2O2 than ABC and ABC undergo contact inhibition without stepping into senescence, an aspect similar to what happen during age-related macular degeneration (Asatryan et al, 2022. Reference 29 in the present paper). Also, we found a higher PLD1 and PLD2 gene expression in ABC cells compared to hRPE49 or ARPE-19 cells, so they seem to be a good RPE cell line for studying classical PLDs roles. We believe that when working with cell lines it is better to corroborate results in different ones. However, whether ABC or ARPE-19 are a better RPE model to use will depend on the aim of the work (see lines 341-351).

Round 2

Reviewer 2 Report

Authors have addressed all the concerns in my previous revision.